# DABS: A Domain-Agnostic Benchmark for Self-Supervised Learning

**Alex Tamkin**[†]
Stanford University

**Vincent Liu**
Stanford University

**Rongfei Lu**
Stanford University

**Daniel Fein**
Stanford University

**Colin Schultz**
Stanford University

**Noah Goodman**
Stanford University

## Abstract

Self-supervised learning algorithms, including BERT and SimCLR, have enabled significant strides in fields like natural language processing, computer vision, and speech processing. However, these algorithms are domain-specific, meaning that new self-supervised learning algorithms must be developed for each new setting, including myriad healthcare, scientific, and multimodal domains. To catalyze progress toward domain-agnostic methods, we introduce DABS: a **D**omain-**A**gnostic **B**enchmark for **S**elf-supervised learning. To perform well on DABS, an algorithm is evaluated on seven diverse domains: natural images, multichannel sensor data, English text, speech recordings, multilingual text, chest x-rays, and images with text descriptions. Each domain contains an unlabeled dataset for pretraining; the model is then is scored based on its downstream performance on a set of labeled tasks in the domain. We also present e-Mix and ShED: two baseline domain-agnostic algorithms; their relatively modest performance demonstrates that significant progress is needed before self-supervised learning is an out-of-the-box solution for arbitrary domains. Code for benchmark datasets and baseline algorithms is available at `https://github.com/alextamkin/dabs`.

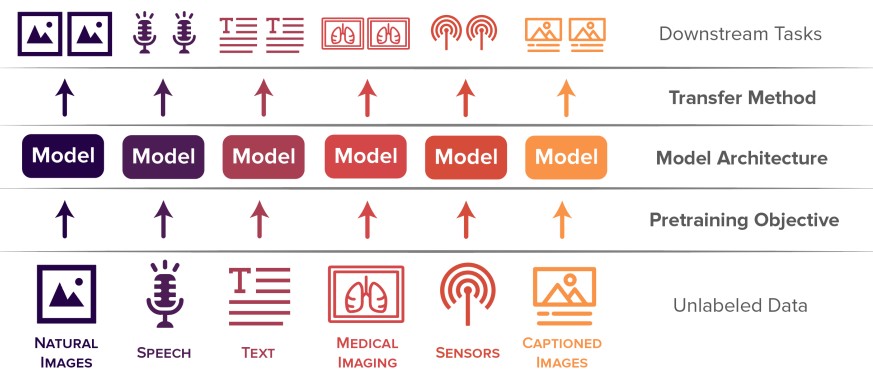

Figure 1: **The DABS Benchmark.** A domain-agnostic self-supervised algorithm consists of 1) a model architecture, 2) an objective used to pretrain the model on unlabeled data, and 3) a transfer method used to deploy it on a downstream task (bolded items). A successful algorithm will achieve high performance on downstream tasks **while holding these components constant across domains**.

---

[†]`atamkin@stanford.edu`

35th Conference on Neural Information Processing Systems (NeurIPS 2021) Track on Datasets and Benchmarks.

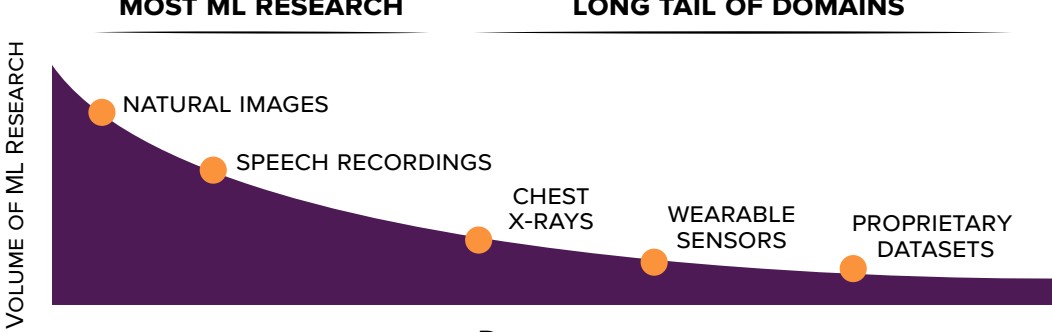

Figure 2: **Domain-agnostic SSL could reduce the need for labeled data across a long tail of domains and application areas.** Currently, developing an SSL algorithm requires considerable domain-specific trial-and-error, limiting it to domains with the most active ML communities. Advances in domain-agnostic methods could make SSL available to all domains, as well as provide scientific insights into principles underlying the success of SSL across modalities. (Figure is illustrative, not based on real data.)

# 1   Introduction and Motivation

Self-supervised learning (SSL) is on the rise across machine learning (ML), with notable recent successes in computer vision [19, 41, 39], natural language processing (NLP), [26, 103, 23] and speech processing [88, 3]. SSL enables a model to acquire useful capabilities from unlabeled data; these capabilities can then be leveraged to drastically reduce the amount of labeled data needed to achieve high performance in a domain—a crucial advance given the time and expense needed to annotate datasets of millions of labels.

However, the potential impact of SSL is arguably greatest outside of the fields where it has currently seen the most success. Medical and scientific domains, for example, are rich in unlabeled data, yet the time and expertise needed for annotation far exceeds that for computer vision or NLP. This means that methods which reduce the need for labeled data are especially impactful in these settings.

Unfortunately, the most popular SSL methods are currently domain-specific—for example, the color jitter distortions used in SimCLR [19] are inapplicable to black-and-white chest x-rays, and the masked language modeling task used in BERT [26] is not directly applicable to spoken language, which is untokenized. Furthermore, these algorithms are challenging to develop, requiring costly trial-and-error by ML experts [19]. Unfortunately, while a great number of domains may benefit from SSL, this distribution exhibits a long tail where the vast majority of domains lack the ML expertise and resources to develop custom SSL solutions.

We argue that an appealing alternative to developing domain-specific SSL methods is to develop domain-agnostic techniques which work across a wide range of settings without extensive modification. Such domain-agnostic SSL algorithms could benefit the field in multiple ways:

1. **Making SSL work out-of-the box.** The most important impact of domain-agnostic SSL would be turning SSL into an out-of-the-box technology capable of being used in any domain of interest without significant ML expertise (Figure 2, right). Aside from medical and scientific domains, this would also benefit the combinatorial number of multimodal settings which currently require novel algorithms to learn the relationships between modalities [61, 20, 86].

2. **Improving handcrafted SSL methods.** Several works have investigated how more general SSL methods can be combined with domain-specific knowledge (e.g. image augmentations) to provide gains [54, 85, 90]. This suggests that advances in domain-agnostic SSL could benefit popular ML domains as well (Figure 2, left), through combination with domain-specific methods.

3. **Uncovering fundamental principles of SSL across domains.** Communities such as computer vision and NLP currently have relatively disjoint investigations into SSL methods; this may obscure common scientific principles underlying the success of algorithms across modalities. Research on domain-agnostic methods may discover these general principles, which could benefit all domains.

However, despite the promise of domain-agnostic SSL, there has been no standardized way to evaluate or drive progress in a cross-cutting way among different communities. To fill this need, we propose DABS, a **D**omain **A**gnostic **B**enchmark for **S**elf-supervised learning. DABS measures how well a single SSL algorithm works on many different domains, as opposed to just one. The benchmark is comprised of seven domains representing different kinds of data: natural images, English text, speech, chest x-rays, multichannel sensor data, multilingual text, and images with text descriptions. Each domain contains one unlabeled dataset for self-supervised learning, and at least one labeled dataset to assess *transfer*: how well the SSL model can adapt its abilities to downstream tasks. Models are assessed by their average transfer learning performance on downstream tasks across domains.

We anticipate a few common questions about DABS:

**Why do we need a benchmark for domain-agnostic SSL?** Benchmarks catalyze progress by providing a common set of tasks, rules, and evaluation criteria for research towards a particular goal. In this case, DABS provides a standardized way to evaluate the performance of domain-agnostic methods. Fixing the choice and preprocessing of datasets allows for clean comparisons over a range of diverse domains, enabling researchers to pinpoint what specific changes contribute to the success of different methods. Furthermore, the provided infrastructure for data processing, training, transfer, and evaluation significantly reduces the barrier to entry for other researchers interested in these questions. Without a low-friction way to evaluate algorithms in a standard way, many researchers may not bother with the significant effort needed to gather and process 25+ different datasets across distinct domains, impeding cumulative progress as a field towards domain-agnostic SSL.

**Why might we expect there to be a good domain-agnostic method?** Many kinds of naturally-occurring and artificial data exhibit structure which humans can exploit to learn transferrable skills [16, 83, 13, 31]. Human-relevant data (as opposed to white noise) is often generated by some complex generative process. For example, the PAMAP2 wearable sensor dataset [74] is produced by a cascade of latent factors including human interpretation of an activity command, the bodily mechanics of the activity's execution, and the physical properties of different kinds of sensors that produce measurements. Domain-agnostic pretraining objectives may enable models to capture these latent factors if they are useful for compressing the data (e.g. via density estimation objectives like language modeling [80]) or distinguishing examples from one another (e.g. via contrastive learning objectives [40]). Furthermore, studies on transfer learning of deep networks suggests there exist useful and general "subroutines" learned by SSL models which enable the model to transfer well to new datasets [104, 84]. Empirically, the recent progress of existing domain-agnostic methods [85, 54, 90] is cause for optimism about the future success of this research direction.

**What does domain-agnostic mean?** The goal of DABS is to catalyze the creation of SSL algorithms which are useful out-of-the-box across different domains. We operationalize this goal by evaluating algorithms on a suite of seven diverse domains crossing many different fields where machine learning is used. We also propose several constraints on submissions, described in Section 3, to prevent "overfitting" to these domains. For example, algorithms must use a set of provided dataloaders and keep their architecture and pretraining objective constant across domains (Section 3). However, we also rely on a degree of pragmatism and collaborative ethos from users of DABS to abide by the spirit of the benchmark; for example, a "domain agnostic" algorithm that uses an if-statement to select domain-specific methods for each domain would likely not generalize to new domains. To this end, we will add new domains in the future as an ultimate test of the generalizability of proposed algorithms.

To summarize, our **contributions** are:

1. We propose and motivate the task of domain-agnostic self-supervised learning.
2. We present a benchmark for measuring domain-agnostic self-supervised learning, including standardized data loaders and rules for ensuring fair comparisons across submissions

3. We present two domain-agnostic baseline algorithms and evaluate them on our benchmark, showing relatively modest improvements over baselines that were not pretrained. This suggests ample room for future methods to drive progress.

## 2 Related Work

**Single-domain transfer learning benchmarks**   Several works have created benchmarks from multiple datasets in a single domain, often with the aim of measuring the general understanding capabilities of a single model by measuring its performance across those tasks. Such datasets have been developed in natural language processing [91, 92], computer vision [87, 108, 107], speech processing [81, 101], molecular machine learning [99], robotics [106], graphs [45], and reinforcement learning [6], among others.

While these datasets often focus on how a single model can adapt to multiple downstream tasks in a domain, they are typically agnostic to the specifics of the pretraining process—encouraging a "no holds barred" setting where larger models, datasets, and domain-specific assumptions are all utilized to increase downstream accuracy. By contrast, our goal here is to develop general techniques that can be used out-of-the-box for acquiring transferrable capabilities from unlabeled data in any domain. Thus, we hold the pretraining data fixed, allowing researchers to improve only the (domain-agnostic) pretraining algorithm, model architecture, and transfer procedure.

**Modality-agnostic architectures**   In order for an SSL method to be usable out-of-the-box, the model architecture must be applicable in new domains without much customization. Transformers [89], originally developed for text, have recently shown promise as a more general architecture for SSL through successful extensions to computer vision [30], molecular data [78, 75], speech processing [38], and multimodal data [61, 86, 20]. These approaches typically use locality assumptions about continuous data (e.g. breaking the input into patches) to map the data into a sequence of embeddings, which are then processed by the transformer. Our baseline algorithms, e-Mix and ShED, leverage similar ideas to train transformer models across all seven of our domains, however we expect and encourage future work to explore other flexible architectures, such as Perceiver [50] which relaxes these locality assumptions at the cost of increased computational demands.

**Domain-agnostic self-supervised algorithms**   Several streams of work have recently developed more general SSL methods. Recent work in contrastive learning has sought to reduce the reliance of the objective on domain-specific augmentation functions. The most common approach seeks to find heuristic augmentations which are applicable across a wider range of domains [54, 90, 105], while other work seeks to develop generative models which learn data-dependent distortions during training with a suitable objective [85]. Outside of contrastive learning, masked language modeling [26] or replaced token detection [23], have been applied to other kinds of tokenized or discrete data [47, 46, 21], but require modification when applied to continuous domains [30, 59]. However, none of these algorithms are applicable out-of-the-box on the DABS datasets, so in this work we propose simple domain-agnostic extensions of algorithms in both of these families.

**Transfer learning methods**   Evaluating a self-supervised learning algorithm requires a method to transfer model's abilities to other tasks of interest [1, 17, 11]. These strategies are typically quite domain-agnostic, but involve tradeoffs between various properties, including complexity, downstream performance, and the degree to which they modify the original model. The two most common transfer strategies have historically been training simple linear classifiers on activations extracted from these pretrained models [29, 64, 69], or finetuning, where one can often achieve higher performance by specializing the entire model to the downstream task via end-to-end training [79, 37, 70, 26]. However, recently, other transfer methods have shown initial success in capturing the benefits of both these extremes, including directly specifying the task in natural language [15], as well as approaches that train only a small subset of parameters in the original model [34, 7] or that inject trainable features into the input [32, 60, 55] or hidden states [57] of the model. We permit and encourage users of DABS to investigate different domain-agnostic transfer methods in order to understand their tradeoffs and performance across different domains.

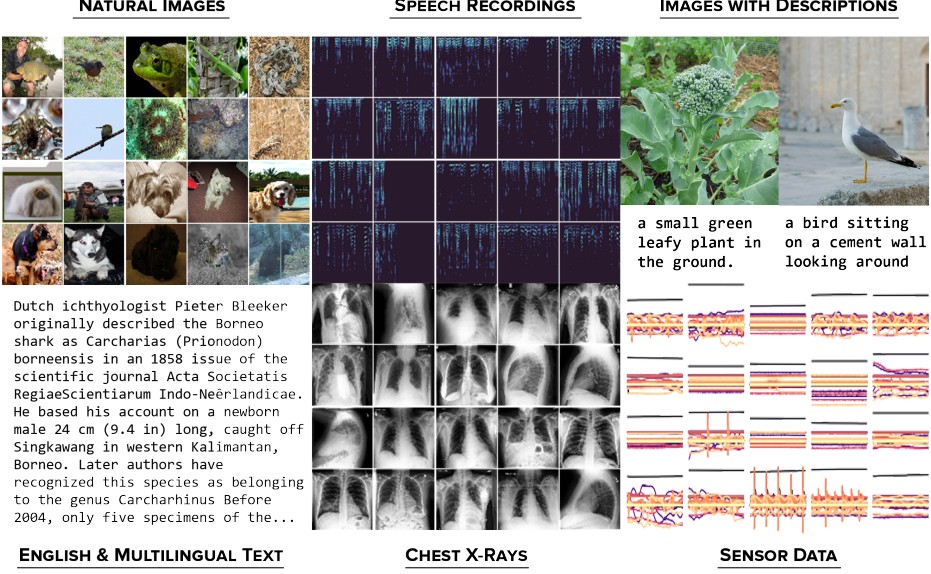

Figure 3: **Different domains and datasets used in this work.** Examples from pretraining datasets of different domains (clockwise from top-left): CIFAR-10, LibriSpeech, MS COCO, PAMAP2, CheXpert, and WikiText-103. For sensor data, each line is a reading from a different sensor.

## 3   Evaluating Domain-Agnostic SSL Algorithms with DABS

How should we evaluate a domain-agnostic SSL method? In DABS, the ultimate goal is to produce a general out-of-the-box solution for SSL across domains—one that generalizes without much modification to arbitrary desired applications. However, one challenge is that SSL methods are comprised of many factors, including the data, pretraining objective, model architecture, and transfer method. Here we describe how the rules of DABS ensure fair comparisons across each of these:

**Datasets**  DABS consists of multiple datasets spread across seven domains (detailed in Section 4). Each domain contains an unlabeled dataset used for pretraining, and one or more labeled datasets to evaluate transfer. These transfer datasets include both labeled subsets of the pretraining dataset as well as different labeled datasets in the same domain. To establish fair comparisons across algorithms, we standardize the data loading process, ensuring the same train/test splits, resolutions, tokenizers, and other details. As our primary aim is to measure the performance of methods in a domain-agnostic setting, as opposed to competing with domain-specific methods, we also prohibit the use of data augmentations which vary between domains (e.g. cropping-and-resizing used in natural images). While this may result in lower performance on transfer metrics relative to domain-specific approaches, past work suggests that many domain-agnostic methods can be combined with domain-specific techniques to provide gains [85, 54, 90].

**Pretraining method**  The goal of an SSL objective is to enable a model to acquire general capabilities from unlabeled data. To evaluate this, a single pretraining method is used to train a model on each pretraining dataset (Figure 1). Crucially, this method may not be changed by hand between modalities (e.g. adding auxiliary losses for text). However, we do allow adaptive methods that alter the pretraining task in a general way based on the model's interaction with the unlabeled data, e.g. by learning a generative model to produce input-dependent distortions [85].

**Model architecture**  While the main model architecture should be kept constant, a key challenge is that different datasets have different input types and dimensions. Some recent works, such as Jaegle et al. [50], attempt to build more data-agnostic model architectures capable of handling different kinds of inputs, however this comes at a compute cost. We take a more permissive stance, allowing

different types of data to have specialized embedding modules that convert an example from the dataset into a series of vectors. These vectors then serve as the input to a model which is otherwise held constant across the datasets.

We provide a starter set of embedding modules compatible with sequence modeling architectures such as transformers [89]. However, we encourage development of other general input modules as long as their tradeoffs are made clear when comparing against other methods—including running ablation studies to isolate the effects of changing the embedding module from the effects of proposed training strategies. We also emphasize that these embedding modules should encode as few assumptions about the data as possible: the goal is to produce a general, out-of-the-box strategy for SSL. For continuous 1D and 2D inputs, we use patch embeddings [30] with standardized patch sizes (see Table 2), and for text we use a standard token embedding lookup table, where the text is tokenized using a standard tokenizer.[2] For multimodal data, we apply the appropriate embedding modules to each input, then concatenate the resulting embedding sequences in the same order each time. These embedding modules are the only parts of the model which differ across domains, enabling the main architecture to operate identically on each embedding sequence.

**Transfer method**   The ultimate measure of an SSL model is how well it performs when its capabilities are adapted to new tasks. Crucially, transfer methods are distinct from pretraining objectives, and must be compared in their own right as first-class components of an SSL algorithm. Like pretraining techniques, transfer methods exhibit tradeoffs beyond task performance: for example, finetuning a model may produce high accuracy, but requires a separate copy of the model for each use case. Other methods, such as linear evaluation [110], in-context learning [15], and p-, prefix-, and prompt tuning [60, 55, 57] enable the same model to be reused across tasks, but may achieve worse performance in some settings. We allow any transfer method as long as it is held constant across domains and downstream tasks.[3]

**Final evaluation metric**   There are many metrics one might use to compare SSL algorithms, including downstream accuracy, speed, fairness, and cost [33]. In this work, we focus on absolute performance of the model on the given data, for a given number of pretraining steps. However, participants may also be interested in other factors, including compute or data efficiency, or the scaling coefficient of techniques [51, 42]. We encourage users of the benchmark to consider any of these, as long as they make clear what previous work is comparable and perform ablations to identify which specific changes impacted the metric being measured.

## 4   Domains and Datasets

Here, we describe the domains and datasets that comprise DABS. Domains were chosen to span a mix of impactful areas, including domains with both large ML research communities (natural images, text, speech) as well as domains where methods are more nascent (medical imaging, sensor recordings, vision-and-language). Dataloading and preprocessing within each dataset has been standardized to ensure fair comparisons; more information about data processing for each modality may be found in the Appendix.

**Natural images**   Two-dimensional color images of the natural world is a deeply-studied domain in machine learning. For an unlabeled pretraining dataset, we use ImageNet [77], a pervasive image classification benchmark in machine learning consisting of 1.3M images from 1000 classes, including meercat, streetcar, and chocolate sauce. We measure the average transfer accuracy on several image recognition tasks commonly used to assess transfer of pretrained vision models [19, 39]: the FGVC-Aircraft dataset [63], the Caltech-UCSD Birds Dataset [96], the German Traffic Sign Recognition Benchmark [43], the Describable Textures Dataset [22, DTD], the VGG Flower Dataset [67], and the CIFAR-10 dataset [53].

---

[2]We use the common HuggingFace BertTokenizer and XLMRobertaTokenizer for English and multilingual text, respectively.

[3]Note that a transfer method does not presuppose a particular loss function, which may in general vary across tasks. For example, one can finetune a model for both regression and classification tasks.

**Speech recordings**    Speech processing is another large community with significant ML presence. We pretrain using the LibriSpeech corpus [68], a large English-language audiobook corpus commonly used for pretraining. We evaluate transfer to several datasets, including the VoxCeleb [66] and LibriSpeech [68] speaker recognition datasets, and the Fluent Speech Commands (action, object, and location classification) [62], Google Speech Commands [94], and AudioMNIST [5] utterance classification tasks. To prepare inputs for models, we preprocess examples into log-mel spectrograms— a format which differs significantly from natural image data, and thus may pose challenges for natural image-specific SSL approaches.

**Monolingual English Text**    The discrete, tokenized nature of text data makes it very different in form from the two previous continuous domains. Historically, monolingual English text has been a dominant focus shaping the development of self-supervised pretraining in NLP [25, 44, 69, 70, 26, 15]. To assess whether domain-agnostic approaches can match the performance of methods tailored to this well-studied domain, we consider pretraining on WikiText-103 [65], a 100-million token English-language dataset collected from the set of *Good* and *Featured* Wikipedia articles. For transfer, we evaluate on the GLUE benchmark [91], a suite of English language tasks including natural language inference, sentiment classification, and paraphrase classification, commonly used to measure transfer of pretrained models.[4]

**Multilingual Text**    Unfortunately, machine learning learning approaches designed with only an individual language in mind are unlikely to perform equally well across the broader range of human languages [9, 10, 8, 76, 98]. To assess the generality of pretraining approaches on multilingual, typologically diverse data, we consider the mC4 dataset [71], a filtered multilingual web crawl corpus.[5] For pretraining, we interleave the mC4 subsets for English, Spanish, French, German, Chinese, Korean, and Japanese, meaning that the fraction of examples seen for each language during pretraining is constant (though the kind and number of unique tokens for each dataset may differ— reflecting the heterogeneity of data availability across languages). For transfer, we evaluate on the PAWS-X tasks [102], a set of seven adversarial paraphrase identification datasets in English, Spanish, French, German, Chinese, Korean, and Japanese.

**Medical imaging**    Medical image understanding encompasses a rich set of domains which often possess ample unlabeled data yet limited labeled data, making them ideal targets for SSL. However, the statistics of medical images can differ significantly from natural images, including lower variation across many inputs and subtler task-relevant features that indicate presence of a pathology [72]. Medical imaging boasts less of an ML presence than natural images, despite the fact that many techniques developed for the former may not apply—e.g. color transformations for black-and-white scans. We focus on chest x-rays as a representative medical imaging domain. We pretrain on the large CheXpert [48] dataset of chest x-rays, and assess how well the pretrained model adapts to binary multiclass classification of five observations: atelectasis, cardiomegaly, consolidation, edema, and pleural effusion.[6] To assess transfer to other chest x-ray datasets, we also measure multiclass binary classification of eight observations in the ChestX-ray8 [93] dataset: atelectasis, cardiomegaly, effusion, infiltration, mass, nodule, pneumonia, pneumothorax.

**Multi-channel sensor data**    Many scientific applications are data-rich and show significant promise for SSL. However, many such domains have a very scarce ML presence compared to domains like natural images. As an example, we consider multi-channel sensor data from wearable devices. We use the PAMAP2 dataset [74], consisting of 52-channel sensor recordings (including accelerometer, gyroscope, and magnetometer data) recorded from different body parts as participants perform varied physical activities. We measure transfer to the labeled PAMAP2 task of human-activity recognition: classifying the activity captured in a given recording snippet (e.g. cycling or walking). Examples are arranged into 1-dimensional time series of 52-channel measurements. Thus, this modality contributes to the benchmark's coverage both in terms of shape and content.

---

[4]The GLUE benchmark tasks are CoLA [95], SST-2 [82], MRPC [28], QQP [49], STS-B [18], MNLI [97, 14], QNLI [73, 91], RTE [24, 4, 36, 12], and WNLI [56].

[5]However, we note that multilingual data is not sufficient to ensure inclusion of multicultural perspectives, and we encourage future work conducting deeper analysis and documentation of mC4, similar to Dodge et al. [27].

[6]These are known as the CheXpert "competition tasks" [48].

**Images with text descriptions** Multimodal models are an increasingly important area in machine learning. An an example domain with two modalities, we consider natural images with paired English-language text descriptions. We pretrain on image-description pairs from the COCO dataset [58]. We then assess the model's ability to adapt to 1) detecting whether an MS COCO image-text pair is matching or mismatched, and 2) the Visual Question Answering task [2], reformulated as a binary task to predict whether a question-answer pair correctly describes an image.

# 5 Domain-Agnostic Baseline Algorithms

A domain agnostic self-supervised algorithm is comprised of a domain-agnostic encoder, pretraining objective, and transfer method for learning downstream tasks. However, to the best of our knowledge no previously-proposed method is compatible off-the-shelf with all of the domains in DABS. To establish some baseline approaches, we propose two simple, domain-agnostic algorithms that we evaluate on DABS, which we describe below and hope will serve as useful starting points for future research on domain-agnostic SSL. The core idea behind these algorithms is simple: use a small set of domain-specific embeddings modules to map inputs into an embedding space, and then define the pretraining task directly on those embeddings as opposed to the original inputs.

## 5.1 Transformer Architecture

Our algorithms use a generalized architecture based on transformers [89]. These transformers take as input the sequence of embeddings obtained from the DABS embedding modules, then process them through a series of self-attention and feed-forward layers. We use a 12-layer transformer with hidden size 256, 8 attention heads, and dropout with probability 0.1. To obtain a feature vector for the input, the activations from the final layer are averaged and projected to a 128-dimensional vector. The sequence lengths vary in length depending on the dataset input dimensions and patch sizes, listed in Table 2. The same Transformer architecture is used across all experiments and is optimized with the AdamW optimizer with learning rate 1e-4 and weight decay 1e-4.

## 5.2 Pretraining Objectives

Given this common architecture, the models are then optimized with respect to a pretraining objective, which enables them to learn useful capabilities and representations from the data. We propose two baseline domain-agnostic SSL objectives, which generalize existing domain-specific methods:

### 5.2.1 e-Mix: A Contrastive Embedding-Mixup Objective

Contrastive learning and other view-matching objectives have made great strides establishing themselves as competitive or even superior alternatives to supervised pretraining in computer vision [100, 19, 41, 39]. Several works have identified the reliance of these algorithms on hand-designed augmentation or "view" functions as a crucial impediment to applying these methods in a more domain-agnostic method. These works either *learn* these view functions from scratch using a generative model [85], or rely on handcrafted augmentations which are more domain-general such as mixup [54, 90]. However, these works make assumptions about the structure of the inputs (e.g. that they are continuous) and use domain-specific encoders, leaving room for a more general solution.

We propose *e-Mix*, a generalization of the i-Mix approach proposed in Lee et al. [54]. In i-Mix, a batch of inputs $x_{1...N}$ is perturbed with mixup noise [109], where examples are additively mixed with other examples in the dataset. This produces mixed inputs $\tilde{x}_{1...N}$, where $\tilde{x}_i = \lambda x_i + (1 - \lambda)x_{\pi(i)}$ for some random permutation $\pi : \{1 \ldots N\} \to \{1 \ldots N\}$ and mixing coefficient $\lambda \sim \text{Uniform}(0.5, 1)$, chosen for each example. Then, the task is to learn an encoder $f$ such that the vector $f(\tilde{x}_i)$ is close to both $f(x_i)$ and $f(x_{\pi(i)})$ in proportion to their respective mixing coefficients. Formally, the loss is:

$$\ell_{\text{i-Mix}}(x, \pi, \lambda) = -\sum_{n=1}^{N} v_{i,n} \log \frac{\exp(\text{sim}(f(x_i), f(\tilde{x}_n))/\tau))}{\sum_{k=1}^{N} \exp(\text{sim}(f(x_i), f(\tilde{x}_k))/\tau)} \tag{1}$$

where sim denotes the cosine similarity, $\tau > 0$ is the temperature, and $v_{i,n}$ is a virtual label given by

$$v_{i,n} = \begin{cases} \lambda, & \text{if } n = i \\ 1 - \lambda, & \text{if } n = \pi(i) \\ 0, & \text{otherwise} \end{cases} \tag{2}$$

The main generalization provided by e-Mix is that the mixup noise is applied to the outputs of the embedding modules (i.e. the patch or token embeddings), as opposed to the inputs directly, which may not in general be continuous. This enables e-Mix to be applied without changes to each of the seven domains in the benchmark. To obtain $f(x_i)$ for a given example $x_i$, we process an input with the transformer, then mean pool the outputs along the sequence length dimension, and finally pass the resulting vector through a fully-connected layer with output size 128.

### 5.2.2 SHED: A Shuffled Embedding Prediction Objective

In contrast to the contrastive objectives that have become common in continous domains such as images and speech, token-level objectives have been more common in natural language processing. Perhaps the most paradigmatic example is masked language modeling [26], where random tokens in the input are either redacted or modified and the goal of the main network is to denoise the input. While this approach has been successfully applied to text, as well as other domains such as images [30] and audio [59], domain-specific modules are still needed to predict the inputs, which may be downsampled, averaged, or otherwise processed to improve performance.

To avoid this domain-specific complexity, we generalize another family of objectives based on the ELECTRA [23] method for pretraining on text. Rather than reconstruct noised tokens as BERT does [26], ELECTRA involves replacing a subset of tokens in the input with substitutes, then training a detector network to predict which tokens were replaced. Substitute tokens can be chosen randomly, or generated by a BERT network. Similar replacement-detection methods have also recently been applied successfully to tabular data [47], suggesting this objective is not text-specific.

We generalize ELECTRA by applying replacements at the embedding level, instead of the level of input tokens. This enables us to apply the method equally across all modalities, without domain-specific adjustments. To perform the replacements, we select 15% of the embedding positions per input, then shuffle those embeddings among each other according to a random permutation. See the Appendix for more details. The task of the network is then to predict which of the embeddings were shuffled; this is instantiated as a binary prediction task performed by passing each output embedding through a fully-connected layer. We call this method ShED: Shuffled Embedding Detection.

### 5.3 Linear Classification

We transfer our trained models to downstream tasks with linear classifiers, a simple approach which enables the same base model to be adapted to many downstream tasks without storing a separate copy of the model for each task. We use the Adam optimizer [52] with learning rate of 1e-4, $\beta_1 = 0.9, \beta_2 = 0.999$ for 100 epochs. We also compare against a randomly-initialized model which has not undergone training, to quantify the gains attributable to pretraining.

### 5.4 Results

We report average metrics by domain in Table 1, and full results for each transfer task in Table 3. Our pretrained models broadly show gains over models that were not pretrained, although the gains are uneven and often quite modest compared to state-of-the-art domain-specific approaches. While the gains from pretraining are clear across transfer tasks in natural images, speech, text, sensors, and medical imaging, pretraining appears to hurt in the multimodal image-text domain, leaving a clear need for future work. Interestingly, the relative gains for these algorithms also seems to reflect their communities of origin: e-Mix performs best on natural images, while ShED performs better on text-based tasks. Investigating the principles underlying these differences is an interesting avenue for future work, as is discovering methods that work better across all domians.

| Pretraining | Natural Images | Text | Speech | Sensors | Med. Imaging | Images & Text |
|---|---|---|---|---|---|---|
| None | 10.1 | 42.3 | 24.9 | 69.8 | 68.1 | **57.5** |
| e-Mix | **27.9** | 44.1 | **41.8** | 79.5 | 72.4 | 48.9 |
| ShED | 20.9 | **48.4** | 36.5 | **88.7** | **74.5** | 54.3 |

Table 1: **Downstream linear classifier performance of baseline domain-agnostic methods across domains.** Reported numbers are average evaluation metrics across transfer tasks within a domain. Metrics are percent accuracy, with the exception of Medical Imaging (average percent AUC across the five pathologies), and two Text tasks: CoLA (Pearson corrleation) and STS-B (Spearman correlation). "None" refers to a randomly-reinitialized model that has not been trained.

## 6 Limitations and Conclusion

DABS also has limitations. For example, a tradeoff exists between keeping a benchmark reasonably compact so it can be run easily and representing the full range of domains one might care about. Our choice of seven diverse domains represents a middle ground, but DABS is also a "living benchmark," and we plan in the future to introduce domains spanning an even broader range of fields, data types, and applications to drive further progress towards domain-general SSL methods.

In addition, DABS does not capture how well domain-agnostic methods can be combined with domain-specific methods in a hybrid manner, which may be of greater relevance to domains like natural images where many domain-specific augmentations have already been developed. This is an important yet challenging-to-frame problem, and we encourage future work in this direction.

We have presented DABS: a Domain-Agnostic Benchmark for Self-Supervised Learning. Algorithms that perform well on DABS may have significant practical impact, unlocking the benefits of pretraining for a wide array of domains without a significant ML presence. We also hope DABS enables researchers to better understand the general principles underlying self-supervised learning across different domains, especially as the technology matures and becomes more broadly deployed.

## Acknowledgments and Funding Disclosures

We would like to thank Shyamal Buch, Jesse Mu, Jared Davis, Daniel Rothchild, and Mike Wu for useful discussions and feedback. AT is supported by an Open Phil AI Fellowship.

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

## A    ShED Permutations

In ShED, embeddings are shuffled with a derangement: a permutation where no element is placed in its original position. To efficiently compute derangements, we restrict ourselves to a set of cyclic derangements $\pi : \{0 \dots N-1\} \to \{0 \dots N-1\}$ where $\pi(i) = i+1 \mod N$. Here, the inputs to this derangement $i \in \{0 \dots N-1\}$ index the set of randomly chosen embeddings to shuffle, which in general may not appear in the same order as in the original sequence.

## B    Compute requirements

All runs were performed on an internal cluster with single Titan X GPUs. Most pretraining jobs required approximately 1 GPU-day, while the transfer jobs ranged from several minutes (e.g. CoLA) to over 1 GPU-day (VQA).

## C    Dataset Licenses

Below we list each dataset's license, as provided either in the paper proposing the dataset or on the dataset website. For datasets where we were unable to find a license, we list "No License."

- **Natural Images:** ImageNet (ImageNet Terms of Access[7]), CIFAR-10 (MIT License), Describable Textures ("This data is made available to the computer vision community for research purposes."[8]), FGVC-Aircraft ("the images are made available exclusively for non-commercial research purposes"[9]), Caltech-UCSD Birds ("Their use is restricted to non-commercial research and educational purposes."[10]), German Traffic Sign Recognition Benchmark (CC0: Public Domain), VGG Flower (GNU General Public License, version 2).

- **Speech:** Librispeech (CC-BY 4.0), VoxCeleb (CC-BY 4.0), AudioMNIST (MIT License), Google Speech (CC-BY 4.0), Fluent Speech Commands (CC BY-NC-ND 4.0)

- **Monolingual English Text:** WikiText103 (CC BY-SA 4.0), COLA ("We expect that research use within the US is legal under fair use"[11]), MNLI (OANC's license), MRPC (No License), QNLI (CC BY-SA 4.0), QQP ("This data is subject to Quora's Terms of Service, allowing for non-commercial use"[12]), RTE (No License), SST2 (CC0: Public Domain), STSB (CC BY-SA 3.0), WNLI (No License)

- **Medical Imaging:** CheXpert (Stanford University School of Medicine CheXpert Dataset Research Use Agreement[13]), ChestX-ray 8 (CC0: Public Domain)

- **Wearable Sensors:** PAMAP2 (CC-BY 4.0)

- **Images & Text:** MSCOCO (CC-BY 4.0), VQA (CC-BY 4.0)

- **Multilingual Text:** mC4 (ODC-BY), PAWS-X("The dataset may be freely used for any purpose"[14])

## D    Origins and Collection of the Datasets in DABS

DABS makes use of a diverse array of kinds of data. Here, we detail to the best of our knowledge how these datasets were collected, including whether consent was explicitly obtained from humans providing the data.

---

[7] https://image-net.org/download
[8] https://www.robots.ox.ac.uk/~vgg/data/dtd/
[9] https://www.robots.ox.ac.uk/~vgg/data/fgvc-aircraft/#ack
[10] http://www.vision.caltech.edu/visipedia/CUB-200-2011.html
[11] https://nyu-mll.github.io/CoLA/
[12] https://www.quora.com/about/tos
[13] https://stanfordmlgroup.github.io/competitions/chexpert/
[14] https://github.com/google-research-datasets/paws/blob/master/LICENSE

- For the Describable Textures Dataset, Cimpoi et al. [22] set forth that "images [were] downloaded from Google and Flickr by entering the attributes and related terms as search queries."

- For VGG flowers, Nilsback and Zisserman [67] note that they gathered "public images from various websites, with some supplementary images from our own photographs."

- For CIFAR-10, Krizhevsky [53] used several search engines, including Google, Flickr, and Altavista to collect images.

- For ImageNet, Russakovsky et al. [77] state that "We collect candidate images from the Internet by querying several image search engines."

- For the Birds dataset, Welinder et al. [96] state that "The images were downloaded from the website Flickr and filtered by workers on Amazon Mechanical Turk."

- For the PAMAP2 dataset, Reiss and Stricker [74] explicitly note that participants consented to having their data be used for scientific purposes.

- For CheXpert, Garbin et al. [35] note "Individual patient consent is waived for de-identified data in compliance with institutional IRB and federal guidelines."

- For ChestX-ray 8, Wang et al. [93] state that the data was retrieved from an NIH collection of radiology reports and images pulled "with IRB approval (OHSRP #5357)" from the Indiana Network for Patient Care. "The images and reports were de-identified automatically and then the automatic de-identification was manually verified."

- For LibriSpeech, Panayotov et al. [68] note that the LibriSpeech corpus is a read speech dataset based on LibriVox's audio books. "The LibriVox project, a volunteer effort, is currently responsible for the creation of approximately 8000 public domain audio books. Most of the recordings are based on texts from Project Gutenberg, also in the public domain."

- For Google Speech Commands, Warden [94] note that "The dataset has 65,000 one-second long utterances of 30 short words, by thousands of different people, contributed by members of the public through the AIY website."

- For VoxCeleb1, Nagrani et al. [66] state that "VoxCeleb contains over 100,000 utterances for 1,251 celebrities, extracted from videos uploaded to YouTube."

- For Fluent Speech Command, Lugosch et al. [62] state that "The data was collected using crowdsourcing. Participants consented to data being released and provided demographic information about themselves."

- For AudioMNIST, Becker et al. [5] state that "All speakers were informed about the intent of the data collection and have given written declarations of consent for their participation prior to their recording session."

- For WikiText-103, Merity et al. [65] note that "The WikiText language modeling dataset is a collection of over 100 million tokens extracted from the set of verified Good and Featured articles on Wikipedia."

- For German Traffic Sign Benchmark, Houben et al. [43] note that "The dataset was created from approx. 10 h of video that were recorded while driving on different road types in Germany during daytime. The sequences were recorded in March, October and November 2010."

- For FGVC-Aircraft, Maji et al. [63] state that "the photographers kindly made available their images for research purposes."

- For MS COCO, Lin et al. [58] state that they collected images from Flickr, and used Amazon Mechanical Turk for crowdsourcing image annotations.

- For VQA, Agrawal et al. [2] note that "We use the 123,287 training and validation images and 81,434 test images from the newly-released Microsoft Common Objects in Context (MS COCO) dataset," as well as crowdsourcing questions and answers through Amazon Mechanical Turk.

- For mC4, Raffel et al. [71] note that the dataset "is generated from 71 Common Crawl dumps."

- For PAWS-X, Yang et al. [102] note that "The PAWS dataset contains 108,463 human-labeled pairs in English, sourced from Quora Question Pairs (QQP) and Wikipedia pages. PAWS-X contains 23,659 human translated PAWS evaluation pairs and 296,406 machine translated training pairs."

## E   PII and Offensive Content

To the best of our knowledge, none of the DABS datasets contains information directly identifying people involved in the creation of the data. However, some kinds of data, most notably x-ray, speech, and wearable sensor data, may contain enough information about a person such that it could be used to identify them given appropriate information from other sources.

It is quite likely that offensive content exists in some of our datasets, including Wikipedia (the source of WikiText-103), LibriVox (the repository of public-domain ebooks from which LibriSpeech was derived), YouTube (the origin of the celebrity voice snippets that comprise VoxCeleb), and especially Common Crawl (the origin of the mC4 dataset).

## F   Potential Negative Impacts

Domain-agnostic self-supervised learning is a very general technology, as a single SSL model can be applied to many different end tasks, and domain-agnosticity means that we must consider potential SSL models across many different domains. Thus, it is challenging to forecaset potential negative impacts with certainty; however, we can delineate two potential kinds of negative impacts from domain-agnostic SSL research:

- Domain-agnostic SSL may be used for a wide range of purposes, including both bad and good actors. As with good actors, domain-agnostic SSL may enable bad actors to create SSL algorithms for particular domains and applications where they were previously unable to do so, magnifying potential threats. Policies and norms, both formal and informal, may be useful tools for furthering positive impacts while preventing negative applications.

- An increased focus on domain-agnostic methods at the expense of domain-specific methods could have negative impacts. For example, the inductive bias afforded by domain-specific assumptions may enable the development of SSL algorithms in domains where less unlabeled data exists, including lower-resource languages. Furthermore, domain-specific knowledge and expertise are crucial tools both in the curation of data for SSL models and in their informed and ethical deployment to end tasks of interest. Thus, while we hope DABS alleviates the need for developing some domain-specific machine learning techniques through trial and error, domain-specific concerns should still guide the use and deployment of SSL.

## G   Additional Reproducibility Details

In this section, we describe additional details regarding the processing and use of each dataset.

### G.1   Images and Descriptions

The datasets in this domain are both based off the MS COCO dataset, which contains images that vary in size on the order of magnitude of ~600 x ~400. We resize all images to 640 x 480, take a center crop of size 480 x 480, and resize to 224 x 224. The resulting image is divided into patches of size 16 x 16 that are passed into the embedding layer. All descriptions are tokenized and padded or truncated to a sequences of 32 tokens.

In order to create the binary classification task for VQA, we create a sequence of tokens the form `<tokenized question>[SEP]<tokenized answer>` and then encode it with the image as during pretraining. `<tokenized answer>` is randomly chosen from the set of incorrect multiple choice answers in VQA 50% of the time, and is left as the correct answer the remaining fraction. The binary classification task is to determine whether the correct or incorrect answer was chosen.

### G.2 Medical Imaging

CheXpert contains x-ray images that vary in size on the order of magnitude of ~320 x ~320. We simply resize all images to 224 x 224. The resulting image is divided into patches of size 16 x 16 that are passed into the embedding layer.

### G.3 Natural Images

All natural images in our datasets consist of images that are 32 x 32, so we do not apply any preprocessing transforms. The resulting image is divided into patches of size 4 x 4 that are passed into the embedding layer.

### G.4 Sensor

Each measurement in PAMAP2 consists of 52 sensor signals from different parts of the body. We first take random subsamples of length 320. The resulting size 52 x 320 examples are then divided into 1-dimensional segments that each contain sensor readings from 5 measurements (a 1D analogue to the patch embeddings proposed by [30]). These segments are ultimately passed into the embedding layer to produce the inputs for the transformer.

### G.5 Speech

For all speech data, we take a random subsegment of 150526 audio samples (at 16kHz), compute its mel-spectrogram with hop length 672 and 224 mel bins, convert from decibels to power scale, and normalize with a fixed mean and standard deviation of the corresponding speech dataset. The spectrum is treated as a single-channel image and divided into patches of size 16 x 16 that are passed into the embedding layer.

### G.6 English Monolingual Text

All text data is tokenized with the Hugging Face BertTokenizer,[15] and the resulting token sequences are padded or truncated to 128 tokens.

### G.7 Multilingual Text

All text data is tokenized with the Hugging Face XLMRobertaTokenizer,[16] and the resulting token sequences are padded or truncated to 128 tokens.

---

[15]`https://huggingface.co/transformers/model_doc/bert.html`
[16]`https://huggingface.co/transformers/model_doc/xlmroberta.html`

| Dataset | Domain | Phase | Examples (train/val) | Patch Size | Dimensions | Batch |
|---|---|---|---|---|---|---|
| CIFAR-10 | Image | Both | 50,000/10,000 | 4 x 4 | 3 x 32 x 32 | 64 |
| Textures | Image | Transfer | 3,760/1,880 | 4 x 4 | 3 x 32 x 32 | 64 |
| Aircrafts | Image | Transfer | 6,667/3,333 | 4 x 4 | 3 x 32 x 32 | 64 |
| Birds | Image | Transfer | 5,994/5,794 | 4 x 4 | 3 x 32 x 32 | 64 |
| Traffic-signs | Image | Transfer | 600/300 | 4 x 4 | 3 x 32 x 32 | 64 |
| Flowers | Image | Transfer | 6,507/1,682 | 4 x 4 | 3 x 32 x 32 | 64 |
| Librispeech | Speech | Both | 145,265/8,251 | 16 x 16 | 224 x 224 | 64 |
| VoxCeleb | Speech | Transfer | 2,148/555 | 16 x 16 | 224 x 224 | 64 |
| Fluent Speech | Speech | Transfer | 26,250/3,793 | 16 x 16 | 224 x 224 | 64 |
| Google Speech | Speech | Transfer | 115,816/11,005 | 16 x 16 | 224 x 224 | 64 |
| AudioMNIST | Speech | Transfer | 24,000/6,000 | 16 x 16 | 224 x 224 | 64 |
| WikiText-103 | Text | Pretrain | 1,165,029/2,461 | — | 128 | 128 |
| CoLA | Text | Transfer | 8,551/1,043 | — | 128 | 128 |
| SST-2 | Text | Transfer | 67,349/872 | — | 128 | 128 |
| MRPC | Text | Transfer | 3,668/408 | — | 128 | 128 |
| QQP | Text | Transfer | 363,846/40,430 | — | 128 | 128 |
| STS-B | Text | Transfer | 5,749/1,500 | — | 128 | 128 |
| MNLI | Text | Transfer | 392,702/19,647 | — | 128 | 128 |
| QNLI | Text | Transfer | 104,743/5,463 | — | 128 | 128 |
| RTE | Text | Transfer | 2,490/277 | — | 128 | 128 |
| WNLI | Text | Transfer | 635/71 | — | 128 | 128 |
| CheXpert | X-Rays | Both | 223,414/234 | 16 x 16 | 3 x 224 x 224 | 64 |
| PAMAP2 | Sensor | Both | 50,000/10,000 | 5 | 52 x 320 | 256 |
| COCO | V+L | Pretrain | 117,266/4,952 | 16 x 16 | 3 x 224 x 224, 32 | 64 |
| VQA | V+L | Transfer | 248,349/121,512 | 16 x 16 | 3 x 224 x 224, 32 | 64 |

Table 2: **Statistics of different pretraining and transfer datasets used in DABS.**

| Dataset | Domain | Metric | None | e-Mix | ShED |
|---------|--------|--------|------|-------|------|
| CIFAR-10 | Images | Accuracy | 24.20 | 39.43 | **39.63** |
| Birds | Images | Accuracy | 1.62 | **3.86** | 2.95 |
| VGG Flower | Images | Accuracy | 9.03 | **25.96** | 13.03 |
| DTD (Textures) | Images | Accuracy | 7.39 | 8.83 | **18.35** |
| GTSRB (Traffic) | Images | Accuracy | 14.33 | **65.07** | 27.51 |
| FGVC-Aircraft | Images | Accuracy | 2.70 | **10.15** | 5.60 |
| LibriSpeech Sp. ID | Speech | Accuracy | 17.12 | **60.18** | 34.77 |
| VoxCeleb Sp. ID | Speech | Accuracy | 0.59 | 2.43 | **2.81** |
| AudioMNIST | Speech | Accuracy | 33.13 | **80.35** | 67.33 |
| Google Speech | Speech | Accuracy | 4.87 | 19.22 | **20.73** |
| Fluent Locations | Speech | Accuracy | **62.09** | 60.93 | 60.24 |
| Fluent Actions | Speech | Accuracy | 26.15 | 29.87 | **30.53** |
| Fluent Objects | Speech | Accuracy | 30.13 | **39.89** | 39.36 |
| COLA | English Text | Pearson Corr. | 0.00 | 8.40 | **19.00** |
| MNLI_Matched | English Text | Accuracy | 35.80 | 37.80 | **43.10** |
| MNLI_Mismatched | English Text | Accuracy | 36.60 | 37.50 | **44.20** |
| MRPC | English Text | Accuracy | 68.40 | 66.20 | **70.10** |
| QNLI | English Text | Accuracy | 57.70 | 57.90 | **65.50** |
| QQP | English Text | Accuracy | 65.10 | 64.30 | **68.60** |
| RTE | English Text | Accuracy | **54.50** | 51.30 | 52.70 |
| SST2 | English Text | Accuracy | 57.00 | 58.10 | **59.30** |
| STSB | English Text | Accuracy | 4.20 | 11.40 | **17.60** |
| WNLI | English Text | Accuracy | 43.60 | **47.90** | 43.60 |
| PAWS-X EN | Multilingual Text | Accuracy | 57.85 | 54.85 | **61.50** |
| PAWS-X FR | Multilingual Text | Accuracy | 57.80 | 55.90 | **60.90** |
| PAWS-X ES | Multilingual Text | Accuracy | 58.55 | 55.50 | **63.65** |
| PAWS-X DE | Multilingual Text | Accuracy | 58.85 | 56.50 | **62.20** |
| PAWS-X ZH | Multilingual Text | Accuracy | 57.35 | 55.35 | **58.55** |
| PAWS-X JP | Multilingual Text | Accuracy | **57.55** | 57.35 | 52.00 |
| PAWS-X KO | Multilingual Text | Accuracy | 58.80 | 57.70 | **60.60** |
| PAMAP2 | Sensor | Accuracy | 69.81 | 79.48 | **88.69** |
| CheXpert | Chest X-Rays | Avg. AUROC | 68.14 | 72.40 | **74.50** |
| ChestX-ray8 | Chest X-Rays | Avg. AUROC | 57.00 | 63.00 | **63.70** |
| VQA | Vision/Language | Accuracy | **57.50** | 48.90 | 54.30 |

Table 3: **Downstream linear classifier performance of baseline domain-agnostic methods across all datasets.** "None" refers to a randomly-initialized model that has not been pretrained.

