# OpenReview forum: "DABS: a Domain-Agnostic Benchmark for Self-Supervised Learning"
_NeurIPS.cc/2021/Track/Datasets_and_Benchmarks/Round1 — NeurIPS 2021 Datasets and Benchmarks Track (Round 1)_

### Official Review · Reviewer_sXos · 2021-06-21
**A benchmark for Self-supervised learning covering six domains/modalities**

**Rating:** 6
**Confidence:** 3
**Correctness:** Yes

**Strengths:**

 - the proposed benchmark may help researchers to investigate the general principles underlying SSL and to evaluate different model architectures, pretraining objectives and adaptation methods
 - the authors provide a good explanation of the task and the motivation

**Weaknesses:**

 - my main complain is that most of these selected datasets are English or western culture-centric (reflecting from labels in the downstream tasks); surely it is good to build a one-for-all solution supporting various domains/modalities, however, building one domain-specific model working for multiple languages or cultures may be a more practical need. Suggest to explain a bit more about the selection criteria (for example, linking the dicussions in Section 2 and the actual datasets used) [The authors added a multilingual text domain]
 - the definition of 'domain' in this paper looks strange to me, especially I am not sure why the authors consider 'paired image and text' as a special domain. See additional feedback item 1. [The authors add further explanation: they use domain to refer datasets with shared structure]

**Additional Feedback:**

* Regarding transfer method (Section 5.3), it would be better to use fine-tune (in contrast to feature extraction) approach.
* The usage of 'domain' in this paper is questionable, Among these six domains, 'text', 'image', 'speech' are usually considered different modalities; 'medical images' and 'natural images' are different domains; and 'paired image and text' seems a special type of annotation? (a similar example may be paired English-German translation)
* how do you feel methods that use domain similarity to select pre-training data can fit with the proposed benchmark
* Line 532, the author name is 'Hao Tan', not 'Hao Hao Tan'

**Clarity:**

- The paper is well written and easy to understand. The only section I suggest to add more details is Section 5.1, it would nice to see how different these input data are and how they are converted into similar outputs

**Documentation:**

- The authors mention that they will provide detailed instructions on Github
- The authors mention that this will be a 'living benchmark', meaning they plan to introduce more domains (and datasets?).

**Relation To Prior Work:**

Yes

**Summary And Contributions:**

This paper describes a benchmark consisting of both unlabelled and labelled data from six domains: natural images, medical images, text, speech, sensor data and paired image and text.

The authors argue a successful SSL (self-supervised learning) algorithm, which covers model architecture, pretraining objective, and adaptaion method, should work well across a wide range of domains.

Two baseline pretraining objectives, inspired by i-Mix and ELECTRA, are evaluated on the proposed benchmark.

The main contribution of this paper is that it formulates the SSL problem and introduce a corresponding benchmark

---

### Official Review · Reviewer_M3vK · 2021-06-29
**A good starting point for the evaluation of domain-agnostic self-supervised learning**

**Rating:** 7
**Confidence:** 3
**Clarity:** The paper is well-written, easy to fo…

**Strengths:**

The proposed benchmark appears to be the first one that aims to provide a general framework to evaluate domain-agnostic algorithms. A unified evaluation framework would be of significant use for the field of domain-agnostic self-supervised learning.

A strong point of the benchmark is that it contains data with very different modalities and domains. Apart from natural images and text, on which there is a huge amount of ML research, it also contains domains that are not as present in ML research, e.g., sensor data from wearable devices.

The benchmark provides a good initial base for the evaluation of domain-agnostic self-supervised learning algorithms. The provided evaluation code appears to be simple to use and easy to extend. This makes it likely that the benchmark would be adopted by other researchers for the evaluation of their algorithms.

**Weaknesses:**

The proposed downstream tasks for evaluation focus on classification settings. A domain-agnostic benchmark would benefit from the inclusion of other types of tasks (e.g., object detection in natural images). This would allow for a more complete picture of an algorithm's general capabilities. It could also help finding limitations of the domain-agnostic framework. For example, certain kinds of downstream tasks might necessarily require a large amount of task-specific additions to the pretrained architecture (not just a simple linear classifier). This would be in conflict with the goal of a domain-agnostic algorithm.
**Edited to add**: The authors address this in their reply to this review. They point out that there is a trade-off "in terms of how much complexity a single benchmark can capture before becoming unwieldy". They explain that they chose to focus on classification and regression tasks as a first step. While I still think that a greater variety of downstream tasks would be beneficial, the current state of the benchmark is a sufficient contribution. I encourage the authors to extend the benchmark with a greater variety of tasks in future installments.

**Additional Feedback:**

While it cannot be expected that the domain-agnostic baselines perform as well as task-specific methods, it might nevertheless be a good idea to include state-of-the-art results on the downstream tasks. This way, a reader could get an impression of how large the difference between the domain-agnostic baselines and the task-specific state-of-the-art is.

As the embedding modules are necessarily domain-specific to a certain degree, it might be helpful to specify certain restrictions on their nature. This way, it could be avoided that highly-engineered domain-specific embedding modules circumvent the goal of a domain-agnostic algorithm.

In appendix B, "Compute requirements", documentation of not only training, but also inference times, as well as memory requirements would be appreciated.

There is a typo in line 221: "different" should probably be "differs".

**Correctness:**

The data for the benchmark is a collection of various datasets that have been widely used in previous research. The combination of unlabeled pretraining datasets and corresponding downstream task datasets is reasonable and clearly motivated.

In some cases, the pretraining dataset is also used for downstream evaluation tasks, which is a questionable decision. For some domains where this is the case, there is still a decent amount of other datasets used for the downstream tasks. However, for the medical imaging and sensor domains, the respective pretraining dataset is the only dataset used for the downstream task (albeit with a different training objective). This makes conclusions about transfer capablilities of an algorithm difficult. The benchmark would benefit a lot if it also contained other datasets in these two domains. **Edited to add**: The reply of the authors to this review alleviates this concern. The authors note that they will include an additional X-ray dataset. For the sensor domain, they explain that there is no other dataset of the same form and argue why the transfer between tasks on the same dataset is still useful.

The chosen evaluation methods are appropriate for the problems in question.

**Documentation:**

As the benchmark contains only datasets that were previously published by other researchers, the authors of this paper are limited to the information provided by the respective original authors of the datasets. The authors provide information about licenses, origin and collection of the datasets as far as this information is available. For each dataset, the relevant papers are cited. Specific information about dataset access can be found in the provided code.

The descriptions in the paper and supplementary material, combined with the provided code, should enable other researchers to reproduce the results stated in the paper.

**Ethics:**

No dedicated ethics review appears to be necessary.

**Relation To Prior Work:**

The discussion of related work is divided into four different topics: Single-domain transfer learning benchmarks, modality-agnostic architectures, domain-agnostic self-supervised algorithms, and transfer learning methods. For each of these, relevant literature is referenced and connections and differences to the author's contributions are clearly explained.

**Summary And Contributions:**

This paper introduces a benchmark for domain-agnostic self-supervised learning. The aim is to evaluate algorithms that are not geared to a specific domain, but rather are intended to work for a diverse array of domains and modalities. To this end, the benchmark contains datasets from six different domains: natural images, text, speech recordings, medical imaging, multichannel sensor data, and paired text and images. An algorithm under evaluation is pretrained on unlabeled datasets from each of the aforementioned domains. Then, its performance is measured on an array of downstream tasks in the corresponding domain. A domain-agnostic algorithm should perform well on all of the domains.

The authors propose several restrictions on evaluated algorithms to ensure that a method is actually domain-agnostic. Data augmentations that vary between domains (e.g., cropping of images) are not allowed. On each of the pretraining datasets the same pretraining method must be used. As data from different domains can come in a variety of different modalities, the authors allow for specialized embedding modules that convert a data sample into a series of vectors. These are the input to the general, domain-agnostic model. The authors propose a set of embedding modules for the domains in the benchmark.

The paper introduces two algorithms intended to serve as baselines for further evaluations. Both contain a transformer architecture, but differ with respect to the pretraining objective. One of them is based on mixup noise, while the other is based on the detection of replacements in the embedding vectors.

For the evaluation of the pretrained models on downstream tasks, the models are combined with a linear classifier. In four of the six domains, the proposed pretraining algorithms lead to significantly better results than a randomly initialized model. The two algorithms appear to be better suited for different domains. In general, the results show that there is much room for improvement and a need of further research.

---

### Official Review · Reviewer_hJWr · 2021-07-06

**Rating:** 7
**Confidence:** 4
**Clarity:** This paper is clear and well written.

**Strengths:**

This paper has a clear motivation and would be of interest to many in the community. Moving towards more general, domain-agnostic measures is an important and challenging direction in the field, and benchmarks that provide standardized ways of comparing methods are an important contribution.

While certainly not exhaustive, the datasets and tasks included in this benchmark provide a good coverage of the space of tasks, spanning vision, language, speech, multimodal and sensor datasets.

Along with the resources the authors will make public, this work introduces and examines two domain-agnostic self-supervised baselines, constituting a good starting point for those interested in the benchmark.

Overall, the paper is very clear and well written.


**Weaknesses:**


1. There are a few design choices that, if better motivated, discussed and studied, could greatly strengthen this paper:

1.1 What motivates the specific choices made for the embedding modules? E.g. why BertTokenizer was chosen as opposed to a BPE tokenizer, or why log-mel spectrograms are used instead of MFCC features? In general, have the authors experimented on different choices to make a principled decision?

1.2. What precise set of criteria was used for including/excluding datasets from this benchmark?

1.3. What is the motivation for using the average of evaluation metrics as the main point of comparison between methods (e.g. Table 1). My concern here is that we are averaging quantities that are not directly comparable (e.g. correlations with accuracies). Could the authors elaborate on why this is the right thing to do here?

2. Tying to point 1.2, I was surprised with some choices for the pre-training datasets, especially CIFAR-10 for natural images and WikiText-103 for natural language. Once concern here is that these datasets are fairly small compared to datasets typically used for pre-training in these fields, and most self-supervised methods perform much better at larger scales. Could the authors elaborate more on why these are good choices?

3. Why is it fair to say this benchmark is domain-agnostic when there are domain-specific components (the embedding modules)? True domain-agnostic models could exist, by, for instance, processing data at the byte-level. While this might still be impractical in the current state of the field of AI, it would be great if authors could be more open and precise about how they are using the term 'domain-specific'.

4. It would be great to see a broader set of tasks in this benchmark, especially those that include tasks where broader interactions between modalities are necessary (e.g. NLVR2 [1], MultimodalQA [2], among many others), or tasks beyond classification, like object detection, semantic segmentation or generation tasks such as language modeling.

5. There are some important points of comparison missing. Currently, it is very hard to access how far away the introduced baselines are from ideal, since there are no explicit comparisons with state-of-the-art methods on individual tasks, or human experiments.

6. Overall, the experimental results from the methods introduced by this work are quite modest (even leading to performance drops compared to not pre-training at all in some cases). While I point this out in this section, I do not believe this to be a particularly strong weaknesses of the work, since it has other contributions besides the baselines. Moreover, the baselines are reasonable methods and valid points of comparison for future work.

[1] Suhr, Alane, et al. "A corpus of natural language for visual reasoning." ACL 2017.
[2] Talmor, Alon, et al. "MultiModalQA: Complex Question Answering over Text, Tables and Images." ICLR 2021.

**Additional Feedback:**

- How will the constraints described in Section 3 (e.g. prohibiting domain-specific data augmentations or loss functions) be enforced?

- In my opinion, Figure 2 is more confusing than enlightening. The current layout is imprecise and implies to readers that 'domain' is a continuous variable without ever properly defining it. I assume the distribution presented is purely illustrative (that is, no actual data was collected for generating this plot). This should be clarified if true, or more details should be added if not.

- Typo on line 300: language


**Correctness:**

With some minor caveats (see concerns above), the evaluation methods and experiment designs are appropriate.

**Documentation:**

To the best of my knowledge, appropriate documentation is included.

**Ethics:**

Since this work mostly builds on existing, public datasets, I believe it does not introduces new ethical concerns that require further discussion.

**Relation To Prior Work:**

To the best of my knowledge, this work clearly discusses how it differs from previous contributions.

**Summary And Contributions:**

This paper introduces a new benchmark, DABS - Domain Agnostic Benchmark for Self-Supervised Learning, created to measure and catalyze progress in domain-agnostic self-supervised learning (SSL) methods. The benchmark is comprised by six different 'domains': natural images, speech, text, medical imaging, sensor data and captioned images. For each domain, a set of unlabeled data is presented, and one or more downstream tasks are used for evaluation. All of these datasets come from existing work, mostly from well established benchmarks. The authors propose a set of rules enforced for comparing different methods. Some examples include ensuring standardized train/test splits, data resolutions and tokenizers; prohibiting domain-specific methods such as domain-specific data augmentation; enforcing that pre-training methods are (mostly) domain-agnostic, with a shared architecture. Finally, this work introduces two baseline methods, e-Mix, an extension of i-Mix where mixup noise is applied to the continuous outputs of the embedding modules; and SHED, a pre-training strategy inspired by ELECTRA where models are tasked to predict which inputs where randomly replaced .These baselines show modest performance across the suite.

Overall, this paper is clear, well motivated and presents interesting findings and assets that will be useful for advancing progress in the field. Despite some minor concerns I list below, this work overall introduces solid contributions, and I recommend its acceptance to the conference.

(Jul 20 update): I have read the rebuttals, along with the other reviews. The author responses, along with the newly introduced modifications/additions, certainly add value to this paper. Overall, while I still have some minor reservations, I believe this paper would be of interest to many, and thus will keep my original rating.

---

### Official Review · Reviewer_TGAw · 2021-07-07
**Interesting benchmark on multi-domain SSL**

**Rating:** 6
**Confidence:** 4
**Correctness:** Yes.
**Clarity:** Yes.

**Strengths:**

1. The motivation of a benchmark to evaluate domain-agnostic SSL method is clearly presented. If there is a successful method that can be applied to multiple domains without significant changes, it would be very valuable.
2. The benchmark covers a wide range of datasets from different domains.
3. Two relatively straightforward baseline algorithms are evaluated. However detailed analysis on the results are missing, only with an aggregated average result.

**Weaknesses:**

1. Using CIFAR as pre-training datasets for natural images seem too restrictive. CIFAR are 32x32 images that are not so natural. Successfully transferring from CIFAR to a reasonably high-resolution dataset seems not realistic. Why not use ImageNet?

2. The benchmark still leaves room for the participants (e.g., in the embedding parts) to freely choose their ways of transforming inputs, which can cause unstandardized comparisons.

3. The motivations of a domain-agnostic SSL argued in the paper seem to equally apply to supervised learning. Why not construct a domain-agnostic supervised learning benchmark first? That can possibly have a bigger impact.


**Additional Feedback:**

See strengths and weaknesses.

**Documentation:**

Yes

**Ethics:**

I can't see any ethical concerns but it's not discussed in paper.

**Relation To Prior Work:**

Yes

**Summary And Contributions:**

The paper proposes a domain-agnostic SSL benchmark suite consisting of six domains. Each domain consists of one unlabeled training dataset and several transfer datasets for evaluation. Embedding modules are used to standardize the input format, and a consistent architecture and training algorithm is enforced in the challenge. Two baseline algorithms adapted from previous works but applied on embeddings are presented.

---

### Comment · Area_Chair_suTx · 2021-07-14
**Reminder to authors: the discussion phase ends on July 14 (anywhere on earth)**

The reviewers provided detailed comments and it would help the review process to know the authors' perspectives on the points raised. For instance, two reviewers mentioned that CIFAR-10 is an uncommon choice as a pre-training dataset.

---

### Decision · Program_Chairs · 2021-07-26

**Decision:**

Accept

**Comment:**

All reviewers vote for accepting the paper. One initial concern was an unusual choice of pre-training dataset (CIFAR-10), but the authors switched to ImageNet for pre-training, which is more common. The authors also extended the range of languages to include non-Western languages, which was another concern brought up by a reviewer. Hence I recommend accepting the paper.